# Label-Free DNA Biosensor Using Modified Reduced Graphene Oxide Platform as a DNA Methylation Assay

**DOI:** 10.3390/ma13214936

**Published:** 2020-11-03

**Authors:** Eliska Sedlackova, Zuzana Bytesnikova, Eliska Birgusova, Pavel Svec, Amir M. Ashrafi, Pedro Estrela, Lukas Richtera

**Affiliations:** 1Department of Chemistry and Biochemistry, Mendel University in Brno, Zemedelska 1665/1, 613 00 Brno, Czech Republic; eliska.sedlackova@mendelu.cz (E.S.); zuzka.bytesnikova@gmail.com (Z.B.); xbirguso@node.mendelu.cz (E.B.); svecpavel@centrum.cz (P.S.); amirmansoor.ashrafi@gmail.com (A.M.A.); 2Central European Institute of Technology, Brno University of Technology, Purkynova 123, 612 00 Brno, Czech Republic; 3Centre for Biosensors, Bioelectronics and Biodevices (C3Bio) and Department of Electronic and Electrical Engineering, University of Bath, Bath BA2 7AY, UK; p.estrela@bath.ac.uk

**Keywords:** nanomaterials, electrochemical biosensor, DNA methylation, rGO, biomedical applications

## Abstract

This work reports the use of modified reduced graphene oxide (rGO) as a platform for a label-free DNA-based electrochemical biosensor as a possible diagnostic tool for a DNA methylation assay. The biosensor sensitivity was enhanced by variously modified rGO. The rGO decorated with three nanoparticles (NPs)—gold (AuNPs), silver (AgNPs), and copper (CuNPs)—was implemented to increase the electrode surface area. Subsequently, the thiolated DNA probe (single-stranded DNA, ssDNA−1) was hybridized with the target DNA sequence (ssDNA-2). After the hybridization, the double-stranded DNA (dsDNA) was methylated by M.SssI methyltransferase (MTase) and then digested via a HpaII endonuclease specific site sequence of CpG (5′-CCGG-3′) islands. For monitoring the MTase activity, differential pulse voltammetry (DPV) was used, whereas the best results were obtained by rGO-AuNPs. This assay is rapid, cost-effective, sensitive, selective, highly specific, and displays a low limit of detection (LOD) of 0.06 U·mL^−1^. Lastly, this study was enriched with the real serum sample, where a 0.19 U·mL^−1^ LOD was achieved. Moreover, the developed biosensor offers excellent potential in future applications in clinical diagnostics, as this approach can be used in the design of other biosensors.

## 1. Introduction

In the last decade, research has confirmed the crucial role of epigenetics in the origin of cancer, its progression, and treatment. Epigenetics play an important role in genetic expressions, which are not affected by changes in DNA sequences. The most significant epigenetic mechanisms are DNA methylation, histone modification, and gene silencing related to RNA. DNA methylation’s essential functions in cells occur under physiological and pathological conditions [1]. DNA methylation is a heritable post-translational covalent modification of DNA catalyzed by DNA methyltransferase (MTase). The process of methylation includes DNA MTase as the carrier of the methyl (−CH_3_) group from S-adenosyl-methionine (SAM), which is a donor to the carbon in the 5th cytosine position in the dinucleotide field of CpG islands. Afterwards, MTase binds the methyl group to carbon in the cytosine’s 5th position to create 5-methylcytosine. The distribution of 5-methylcytosine in the genome is specific for each cell type and is established during embryonic development. Nonetheless, the methylation of DNA is not stable; it may be enhanced by de novo methylation in the cells [2,3]. Compared to normal cells, tumor cells have a disrupted DNA methylation pattern, either reduced (hypomethylation) or increased (hypermethylation) by the number of methyl groups. Hypomethylation usually refers to repetitive DNA sequences, such as long scattered core elements, while hypermethylation refers only to CpG islands. Unfortunately, the mechanism of de novo methylation in normal and tumor cells is not well-understood [4]. The accumulation of DNA methylation changes can occur within a few years before the onset of malignant growth, so these changes could potentially be used in the early diagnosis of cancer [5]. Several methods suitable for DNA methylation determination, such as polymerase chain reaction (PCR), electrochemiluminescence, high-performance liquid chromatography, fluorescence, and colorimetric methods are widely used. These methods are highly effective; however, they have some drawbacks, including expensive instrumentation with a need for a considerable volume of samples, false-positive results, several washing steps, multiple-step samples preparation procedures, and skilled operators [6]. By contrast, biosensors are generally sensitive devices that convert a particular physical or chemical signal through a transducer into a more measurable signal and contain the recognition element composed of biological material (bioreceptor). Biosensors do not require skilled operators, expensive instrumentation, or multiple-step procedures of sample preparation. DNA biosensors, where DNA acts as a bioreceptor, are based on hybridization to a complementary DNA sequence. These biosensors are composed of an immobilized single-stranded DNA (ssDNA) to detect a complementary DNA sequence through hybridization. This leads to the formation of double-stranded DNA (dsDNA). DNA bioreceptors are simple, inexpensive, and widely used to detect DNA methylation [7]. Nevertheless, the development of innovative DNA-based biosensor strategies are highly desirable due to their sensitivity, low limit of detection, cost-effectiveness and specific analysis [8,9]. Unfortunately, in biosensor technology some obstacles must be overcome and improved, such as their stability and their implementation into integrated microfluidic devices [10].

Graphene oxide (GO) is commonly used in many scientific areas and disciplines due to its remarkable features. The specific surface with a large area and oxygen-rich functional groups supports thiolated DNA immobilization, excellent electrical conductivity, and cost-effective synthesis, which makes GO attractive for various electrochemical applications [11]. Reduction of GO is necessary for electrochemistry because the removal of oxygen-rich functional groups reinstates the π-conjugated system and thereby essential conductivity. The reduction of GO by Na[BH_4_] is strong, effective, and results in a similar C/O ratio as reduction by hydrazine [12]. Moreover, GO is widely used in biosensors fabrication due to the unique properties, such as the ratio of its surface area to volume, which ultimately makes the nanomaterial highly sensitive. Nevertheless, Na[BH4] reduction creates a material with smaller sheet resistance, making it more convenient for electrochemical applications [13]. The most important is nanoparticles’ ability to conduct the electrical current, which depends on their distribution on reduced graphene oxide (rGO). Moreover, rGO has excellent thermal and electrical conductivity, large surface area, and additionally exhibit flexible and inert properties. rGO shows significantly reduced nanotoxicity compared to graphene and GO, which allows its wide use in biotechnological and biomedical applications. Among others, metal nanoparticles (NPs) can provide the highest electrical conductivity. The integration of different nanomaterials in the form of a composite has attracted biosensing attention as it can bring about the synergic effect of nanomaterials. Moreover, using this nanomaterial can increase the sensitivity and stability of this assay. Therefore, the nanocomposites, including rGO-silver NPs (AuNPs), rGO-gold NPs (AgNPs), and rGO-copper NPs (CuNPs), were synthesized and employed in a biosensor effect investigation on its analytical figures of merit [14].

In this study, the effect of three different nanocomposites rGO-AuNPs, rGO-AgNPs, rGO-CuNPs on the immobilization process’s stability in signal enhancement was investigated. Many studies used only AuNPs [15,16] for biosensors development. Besides AuNPs, there exist other nanoparticles, such as AgNPs and CuNPs with quite high affinity to (−SH) suitable for biosensor fabrication as well. For instance, AgNPs are cheaper than gold, sufficiently stable, biocompatible, and can even prevent air oxidation [17]. On the other hand, CuNPs can be used as functional biological probes, and they can be quickly and easily prepared, with high deficiency, with almost no toxicity from a biological point of view and can also be intercalated into major DNA grooves [18]. The use of nanocomposites in biosensors fabrication has excellent potential. This is mainly due to the unique properties of used materials, which can replace conventional working electrodes (WEs) [19]. Moreover, the potential for these nanomaterials’ production exists in almost every laboratory, which is very desirable in producing lab-on-chips or biochips, which are expected to be used in a future extension of the proposed study.

## 2. Materials and Methods

### 2.1. Reagents

Reagents in this publication acquired from Sigma-Aldrich (St. Louis, MO, USA) were of analytical grade or better quality unless otherwise stated. The specific synthetics sequences of DNA oligonucleotides and enzymes used in this study were adapted from [20]. Other chemicals used in this work, including methylene blue (MB), 1-mercapto-6-hexanol (MCH), chemicals for buffers preparation: sodium chloride (NaCl), magnesium chloride (MgCl_2_), Trizma^®^ base, hydrochloric acid HCl (37%), sodium hydroxide (NaOH), potassium sulfate (K_2_SO_4_), potassium hexacyanoferrate(III) (K_3_[Fe(CN)_6_]), potassium hexacyanoferrate(II) (K_4_[Fe(CN)_6_]), sodium dihydrogen phosphate (NaH_2_PO_4_) and sodium phosphate dibasic (Na_2_HPO_4_). Reagents for the synthesis of GO and rGO-AuNPs: 0.24 mM tetrachloroauric acid (H[AuCl_4_]), 37% HCl, 0.085 M sodium citrate (Na_3_C_6_H_5_O_7_), 5% sodium carbonate (Na_2_CO_3_), potassium permanganate (KMnO_4_), sodium tetrahydridoborate (Na[BH_4_]) and hydrogen peroxide (H_2_O_2_). Reagents for the of rGO-AgNPs: 10 mM solution of silver nitrate (AgNO_3_) and Na[BH_4_]. Reagents for synthesizing rGO-CuNPs synthesis: copper acetate Cu(CH_3_COO)_2_ and Na[BH_4_]. Moreover, this study is enriched by real human sera samples (ERM^®^ Certified Reference Material, St. Louis, MO, USA). All experiments used ultrapure Milli-Q (Millipore System Inc., Billerica, MA, USA) water with the corresponding resistivity of 18.2 MΩ·cm.

### 2.2. Apparatus

#### 2.2.1. Physical and Chemical Characterization

##### Scanning Electron Microscopy (SEM)

The sample was applied on a silica wafer and dried at ambient temperature (24 °C). This wafer was adhered to by a carbon tape to the stub, which was inserted into the scanning electron microscope (SEM). In this study, the SEM TESCAN MAIA 3 device (TESCAN Ltd., Brno, Czech Republic) coupled with a field-emission electron gun (TESCAN Ltd., Brno, Czech Republic) to investigate the composition and morphology of used nanocomposites. The method was adapted from a previous study [21] with slight modifications.

##### SEM-Energy Dispersive X-ray Spectroscopy (EDX)

In the study, SEM-energy dispersive X-ray spectroscopy (EDX) elemental mapping was performed for the determination of selected elements, such as C, O, Au, Ag and Cu, processed by C Kα1 and 2; O Kα1, Ag Lα1, Cu Lα1 and 2, and Au Mα1 edges, respectively. As a measuring device MIRA 2 SEM (TESCAN Ltd., Brno, Czech Republic) coupled with an EDX detector X-MAX 50 (Oxford instruments plc, Abingdon, UK) was used and the images were evaluated by software AZtec (Oxford Instruments, Abingdon, UK). An external detector SE (Everhart-Thornley, TESCAN Ltd. Brno, Czech Republic) was selected for image processing with an accelerating voltage 15 kV. EDX mappings of selected elements, such as C, O, Au, Ag and Cu were processed using C Kα1 and 2; O Kα1, Ag Lα1, Cu Lα1 and 2, and Au Mα1 edges, respectively. Parameters of measurements were the following: working distance 15.4 mm, input energy 20,000 cts, output energy 16,000 cts with fluctuated dead time around 18–20%. Each analysis took place in 20 min.

#### 2.2.2. Electrochemical Characterization

Electrochemical studies, including electrochemical impedance spectroscopy (EIS) and differential pulse voltammetry (DPV) were performed in a standard three-electrode cell, with an Ag/AgCl reference electrode linked by a salt bridge (50 mM PB + 0.1 M K_2_SO_4_) at constant potential and with Pt auxiliary electrode. A gold electrode with radius of gold disk 1.6 mm (BASi^®^, West Lafayette, IN, USA) was selected as working electrode (WE). An Autolab PGSTAT302N, in combination with a FRA2 potentiostat module (Metrohm, Herisau, Switzerland), was used for taking the electrochemical measurement.

### 2.3. Methods

#### 2.3.1. Synthesis and Reduction of Graphene Oxide (GO)

GO was prepared by the well-established and optimized Hummer’s method with slight modification [22].

The pH value of GO was adjusted to the value 9–10 by the addition of Na_2_CO_3_. The mixture was heated up to 80 °C and afterwards to it was added 800 mg of Na[BH_4_], which is a strong reducing reagent. The temperature was maintained at 80 °C for 1 h. Finally, rGO was collected and rinsed three times with Milli-Q water [23].

#### 2.3.2. Synthesis of Nanocomposites

##### Synthesis of Reduced GO (rGO)-Silver Nanoparticles (AgNPs) and rGO-Copper Nanoparticles (CuNPs)

A volume of 1 mL of rGO solution (5 mg·mL^–1^) was added dropwise to the AgNO_3_ solution and vigorously stirred at 400 rpm. Then, 40 mg of Na[BH4] was slowly added to the mixture, and the final solution was stirred for 24 h to complete reduction. The final nanocomposite was rinsed three times with Milli-Q water, and the final volume was adjusted to 10 mL. For rGO-CuNPs synthesis, the AgNO_3_ was only replaced with Cu(CH_3_COO)_2_.

##### Synthesis of rGO-Gold Nanoparticles (AuNPs)

An amount of 3.75 mg of rGO was added to 50 mL H[AuCl_4_], and the mixture was sonicated for 30 min. Subsequently, the mixture was heated up to 80 °C (stirred at 400 rpm), and 940 µL of Na_3_C_6_H_5_O_7_ solution was added drop by drop. The solution was sustained at 80 °C for 1 h and then cooled down to room temperature [24].

#### 2.3.3. Working Electrode (WE) Preparation

Following the procedure of Keighley et al. [25] the WEs were cleaned and polished with a slight modification. WEs were firstly sonicated in absolute ethanol and then polished with diamond suspension (0.5 µm) (Nanoshel LLC, Wilmington, DE, USA). Furthermore, a Piranha solution was prepared (75% H_2_SO_4_: 25% H_2_O_2_, equals 3:1 volume), where the electrodes were rinsed to remove all the organic impurities. After mechanical and chemical cleaning, WEs were cleaned electrochemically via cyclic voltammetry (CV) in 0.5 M H_2_SO_4_ by potential scanning between the gold reduction and oxidation vs. reference electrode Ag/AgCl for 50 cycles until the voltammogram was established. After that, the WEs were rinsed with Milli-Q water, dried in argon steam (argon 5.0, Messer Industries GmbH, Bad Soden, Germany), then caped and were ready to use.

#### 2.3.4. rGO Immobilization, Single-Stranded DNA (ssDNA-1) Immobilization, and Hybridization with ssDNA-2

In this step, WEs were prepared by dropping 5 µl of the selected nanocomposite (rGO-AuNPs, rGO-AgNPs, and rGO-CuNPs) on the electrode surface. Afterwards, the electrodes were dried for 30 min under an infrared heat lamp (Beurer IL35, Ulm, Germany) with a power consumption of 150 W. Then, the WEs were incubated with a thiol-modified DNA probe, which has a great affinity to attach to the metal surface through the thiol bond with selected nanocomposites. A further step involved incubation with MCH solution overnight at +4 °C while the self-assembled monolayer (S-adenosyl-methionine, SAM) was formed to block the non-specific binding, prevents clusters forming, and helps to anchor the target molecules. Prior to hybridization, we performed back-filling with 1 μM MCH for 90 min. After that, the ssDNA-1 was hybridized with ssDNA-2 in the hybridization buffer [25]. Moreover, the probe and target were incubated at 37 °C in an oven for 1 h to provide full hybridization.

#### 2.3.5. Double-Stranded DNA (dsDNA) Methylation Process and the Digestion of dsDNA by HpaII Endonuclease

The modified WEs with ssDNA-1/dsDNA-2/rGO-AuNPs; -CuNPs or -AgNPs was treated by various concentrations prepared from 1.6 µM stock solution of M.SssI MTase (5; 15; 25; 50; 100; 150; 200; 300; 400; 500 U·mL^−1^ New England Biolabs, Ipswich, MA, USA). The modified WEs were exposed to the M.SssI MTase solution for 2 h at 37 °C in the oven to be methylated and then incubated in 20 mM MB solution for 30 min at ambient temperature. After the successful methylation of the dsDNA, the modified WEs (ssDNA-1/dsDNA-2/rGO-AuNPs; -CuNPs or -AgNPs/M.SssI) were immersed in 20 U·ml^−1^ HpaII in Tango buffer for 2 h at 37 °C in the oven. Then, the WEs were rinsed thoroughly with ultrapure water.

The prepared bare and modified WEs were characterized by using EIS. The impedance spectra were recorded in the solution of 5 mM K_4_[Fe(CN)_6_] + 5 mM K_3_[Fe(CN)_6_] in 100 mM PB + 100 mM K_2_SO_4_ at pH 7.0. EIS was performed between working and auxiliary electrodes with the following parameters: the frequency ranges from 100 kHz to 100 mHz with an amplitude of 10 mV alternating current (a.c.) voltage and 250 mV direct current (d.c.) without external bias vs. reference electrode responds to the formal potential of the redox couple.

#### 2.3.6. Monitoring of M.SssI Methyltransferase (MTase) Activity

The activity of M.SssI MTase was monitored by differential pulse voltammetry (DPV). The potential was swept from 0.0 V to −0.6 V, the potential amplitude was 0.025 V, and the scan-rate was calculated as 0.01 V∙s^−1^. Moreover, 20 mM MB solution was used as an electrochemical probe to measure the quantity of 10 nM target sequences. The modified WEs were incubated for 30 min.

## 3. Results and Discussion

### 3.1. Biosensor Fabrication

Figure 1 presents a simple, label-free, reliable, sensitive, and specific electrochemical DNA biosensor as a DNA methylation assay. In this approach, we have compared three different nanocomposites rGO-AuNPs, rGO-AgNPs, rGO-CuNPs, and their effect on the stability of the immobilization process, and the signal enhancement was investigated. The probe containing thiol groups (–SH) was attached to a layer from the nanomaterials, as mentioned above, by a covalent bond with sulfur. To ensure stability, MCH was used to create the SAM which prevents cluster forming and non-specific binding of DNA. Therefore, the SAM helps to improve the stability of DNA immobilization by proper DNA arrangement [26]. The changes in the electrochemical features of the interfacial electrode surface due to the modifications were characterized via EIS. The methylation assay is based on the DPV response, which is correlated to the amount of the intercalated MB. MB creates a positive charge, leading to electrostatic interactions with negatively charged phosphoric acid of DNA, which allows creating π–π interactions. The transfer of electrons flow from an electrode to the MB, and the obtained signals indicate the quantity of methylated cytosine.

### 3.2. Scanning Electron Microscopy (SEM) Characterization of rGO Decorated with Au, Ag and CuNPs

The images obtained from SEM (Figure 2) display the typical morphology of the specific nanomaterial and declare the presence of the desired carbon nanocomposites. GO synthetized by modified Hummer’s method exhibited a uniform smooth microstructure with fine wrinkles. The reduction process of GO has resulted in a heavily wrinkled material with a preserved microstructure. Both materials exist in the form of large sheets. SEM images reveal the morphological characteristics of the prepared nanocomposites. In Figure 2A, it is evident that the layer of GO is visible and deeper wrinkles indicate the GO was also reduced (Figure 2B) during the synthesis of nanocomposites by Na[BH_4_].

In all nanocomposites, NPs are consistently distributed on the surface of rGO, respectively. The AuNPs in Figure 3C are significantly smaller than AgNPs (A) and CuNPs (B), respectively, whereas all of the nanocomposites are shown in 1 µm scale. AuNPs are distributed in nanometers. The diameter of AuNPs was 9–10 nm. On the other hand, the diameter of AgNPs was approximately 20 nm and the size of CuNPs was approximately 70–80 nm. SEM-EDX mapping image confirmed the occurrence of functional groups and the presence of expected elements that constitute NPs. All these nanocomposites were further characterized by FTIR technique, and obtained data are recorded in the Appendix A. 

### 3.3. Electrochemical Studies of Used Nanocomposites as Modification by Electrochemical Impedance Spectroscopy (EIS)

EIS was used to characterize the successful modification and treatment of the WEs surface. As shown in Figure 4, the impedance spectra are composed of two parts, a semi-portion and a straight line. The semi-portion diameter is a demonstration of the charge transfer resistance, while the straight line represents the Warburg impedance, which is dominant at low frequencies due to diffusion processes [27]. Figure 4 shows the differences between each step of electrode surface modification for the three nanocomposites (rGO-AuNPs, rGO-AgNPs, and rGO-CuNPs). The bare working electrode has shown only a small semicircle signal in every record due to the free-electron transfer process. The equivalent circuit (Figure 4D) describing the system’s behavior and the measured data are interpolated by the curve described by the equation corresponding to the stated circuit model.

The immobilization of rGO-AuNPs with ssDNA-1 increased charge transfer resistance due to electron transfer from the electrode surface. The hybridization of ssDNA-1 with the target DNA further increased the charge transfer resistance. Considering that the utilized electrochemical [Fe(CN)_6_]^3−/4−^ redox pair has a negative charge, the presence of DNA, which is also negatively charged, results in electrostatic repulsion. Hence, the increase of resistance in charge transfer directly after immobilization of the probe and the target DNA is understandable. After treating dsDNA with M.SssI Mtase that causes the methylation, the charge transfer resistance increased again. Increased steric hindrance can be caused by limiting the electrochemical probe’s access to the electrode surface. The addition of HpaII causes dsDNA cleavage in a specific site, and consequently both the steric hindrance and the extent of the electrostatic repulsion decrease. Thus, the sensitivity of the developed biosensor depends on the difference in impedance magnitude in the process of probe immobilization and hybridization with target DNA where rGO-AuNPs show the highest difference because of better affinity −SH groups of ssDNA-1. In addition, modification with M. SssI and Hpall causes high impedance in rGO-AuNPs composite indicates better sensitivity.

### 3.4. Monitoring of Methyltransferase Activity by Differential Pulse Voltammetry (DPV)

As can be seen in Figure 5A, the peak height increases with concentration magnification of M. SssI Mtase concentration. This can be attributed to the higher amount of methylation on cytosine. Figure 5B shows the good linear correlation between the M.SssI concentration and peak current amplitude observed in a broad linear-dynamic range of 5–500 U·mL^−1^. For more reliable results there are displayed two calibration curves in Figure 5B. The equation for the M.SssI in concentration range 5–25 U·mL^−1^ is: *I* (µA) = 0.0024*c* (U·mL^−1^) + 0.029 with the correlation coefficient (R^2^ = 0.9954). The lower limit of detection (LOD) was estimated to be 0.04 U·mL^−1^, and the limit of quantification (LOQ) was 0.13 U·mL^−1^, calculated based on 3*σ*/*k* and 10*σ*/*k*, respectively. In this equation where σ presents the relative standard deviation (RSD%) obtained from three measurements of 500 U·mL^−1^ concentration of M.SssI MTase and *k* is the calibration curve slope. On the other hand, the remaining concentrations (50–500 U·mL^−1^) employ the equation: *I* (µA) = 0.0004*c* (U·mL^−1^) + 0.0984 with the correlation coefficient (R^2^ = 0.99647). The LOD was calculated to be 0.06 U·mL^−1^, and LOQ was 0.19 U·mL^−1^. In the case of ssDNA-1/dsDNA-2/rGO-AgNPs the linear equation was *I* (µA) = 0. 0004*c* (U·mL^−1^) + 0.0995 with R^2^ = 0.9698. The same series of measurements was performed with ssDNA-1/dsDNA-2/rGO-CuNPs, where the equation of concentration dependence of the peak current was *I* (µA) = 0.0003*c* (U·mL^−1^) + 0.0812 (R^2^ =0.9895). Based on the results obtained, ssDNA-1/dsDNA-2/rGO-AuNPs shows better sensitivity in both EIS and DPV measurements.

Nowadays, several studies aimed at a DNA methylation assay have been performed by many scientific groups. Table 1 summarizes the nanocomposites platforms, target enzymes or labels and the LODs used in numerous scientific works. The table is also enriched with the results of our DNA methylation assay.

### 3.5. Demonstration of DNA Methylation Effect in Physiological Sera Samples

Commercial human serum samples were used for real sample analysis (Figure 6). The undiluted serum samples were injected with different concentrations of the M.SssI MTase (Figure 6A). Then, a fabricated biosensor was utilized for monitoring the DNA methylation. As shown in Figure 6B, linear correlation can be observed between the resulting peak current and the desired concentration (*I* (µA) = 0.002*c* (U·mL^−1^) + 0.0263, R^2^ = 0.9934) in a concentration range of the M.SssI from 5 to 25 U·mL^−1^. The LOD for 50 to 500 U·mL^−1^ concentration was calculated to be 0.04 U·mL^−1^, and the LOQ was 0.14 U·mL^−1^. In a 50–500 U·mL^−1^ concentration range of the M.SssI was obtained calibration equitation *I* (µA) = 0.0003*c* (U·mL^−1^) + 0.082 with R^2^ = 0.9959. The LOD was evaluated to be 0.07 U·mL^−1^, and the LOQ was 0.21 U·mL^−1^. The calculation was based on 3*σ*/*k* where σ presents the RSD% of three measurements of the 500 U·mL^−1^ same concentration M.SssI MTase, and *k* is the slope of the calibration curve. The obtained results indicate the applicability of the developed biosensor in complex physiological matrices.

### 3.6. Reproducibility, Repeatability and Stability Studies of the Fabricated Biosensor

Reproducibility, stability, and selectivity studies are an important part of assay methods. The assay’s repeatability was investigated by plotting three independent calibration curves by the same modified electrodes with rGO-AuNPs and calculating the RSD% of the slopes. The calculated RSD% = 4.35 shows good repeatability. This indicates that our approach can be used as an effective M.SssI MTase activity assay.

On the other hand, the biosensor’s reproducibility was studied by analysis of a solution of 20 U·mL^−1^ of HpaII with nine freshly prepared biosensors and then the RSD% was calculated to be 4.78 that confirms good reproducibility of the biosensor. This may stem from the stability of the complex between the rGO-AuNPs composite and the biological compounds.

As a last step the stability of the developed biosensor was investigated as well. The freshly fabricated biosensor was placed in the refrigerator for 7 days at 4 °C and applied for the analysis of the same concentration of DNA MTase, and 93.42% of the initial current response was obtained. This result has confirmed very good stability of the developed biosensor. The biosensor can be stored for a relatively long time without significantly affecting the accuracy of the measurement and it is capable of efficient and reliable analysis over time.

## 4. Conclusions

In conclusion, a simple, sensitive, and reliable electrochemical biosensor for methylated DNA determination and methyltransferase activity was developed. Three different nanocomposites, namely, rGO-AuNPs, rGO-AgNPs, rGO-CuNPs were synthesized, characterized, and applied in the sensor’s construction. The best results in terms of sensitivity were obtained with rGO-AuNPs, which could be due to the high electrical conductivity of AuNPs. Furthermore, the AuNPs offer a good substrate for immobilization of the DNA, which further results in better stability and, consequently, sensitivity of the developed biosensor. This material was the most suitable for the DNA methylation assay due to the broad linear range of M.SssI concentration (50–500 U·mL^−1^). The biosensor developed with rGO-AuNPs showed good LOD (0.06 U·mL^−1^) and LOQ (0.19 U·mL^−1^). The LOD (0.04 U·mL^−1^) and LOQ (0.13 U·mL^−1^) values calculated from the M.SssI (5–25 U·mL^−1^) concentration range indicate that very low concentrations can be reliably detected.

In comparison with other studies in Table 1, this material is appropriate for biosensor fabrication. In the case of analysis in human serum samples, it has been shown that even in the biological matrix, the biosensor can achieve 0.07 U·mL^−1^ LOD and LOQ of 0.21 U·mL^−1^, which is almost identical to the use in analysis in a buffer. The validation of the developed biosensor confirmed its accuracy and precision. Therefore, the developed biosensor offers an alternative that can be utilized in clinical analysis for preliminary measurement.

## Figures and Tables

**Figure 1 materials-13-04936-f001:**
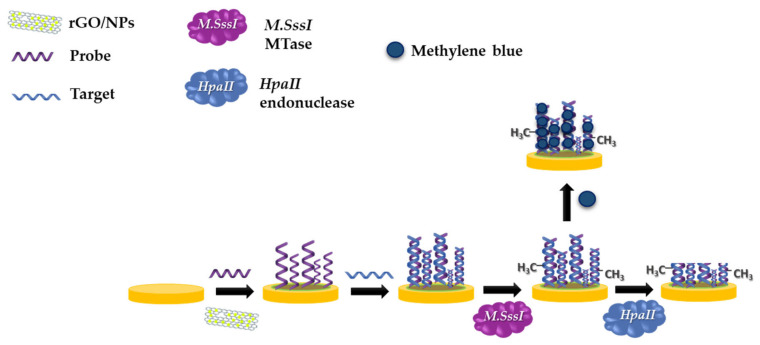
Schematic of biosensor development including each step of immobilization and treatment by enzymes.

**Figure 2 materials-13-04936-f002:**
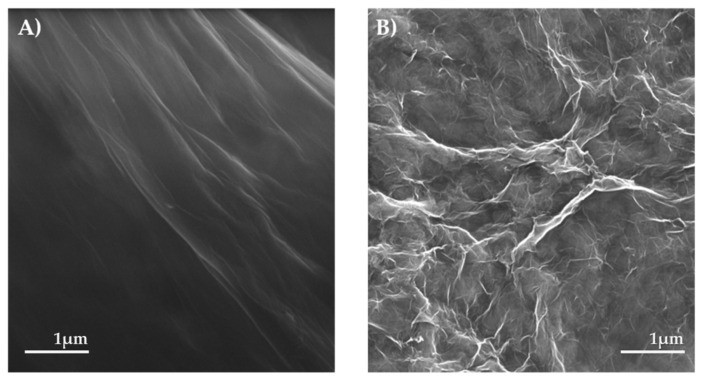
Images obtained from scanning electron microscope (SEM). (**A**) Graphene oxide (GO) and (**B**) reduced graphene oxide (rGO) with 1 µm scale bar.

**Figure 3 materials-13-04936-f003:**
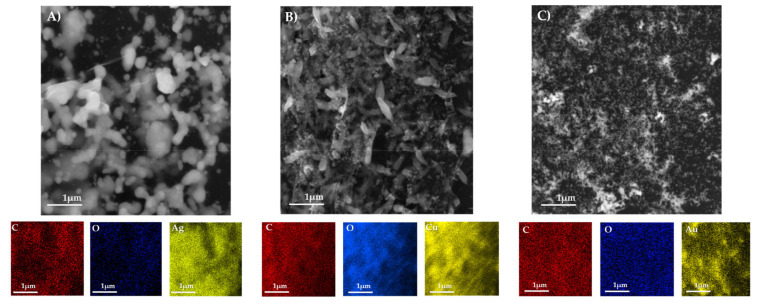
SEM images of nanocomposites of rGO decorated with the different metal nanoparticles. (**A**) silver nanoparticles (AgNPs), (**B**) copper nanoparticles (CuNPs), and (**C**) gold nanoparticles (AuNPs). Below are shown SEM-energy dispersive X-ray spectroscopy (EDX) images from the elemental mapping of demanded nanocomposites.

**Figure 4 materials-13-04936-f004:**
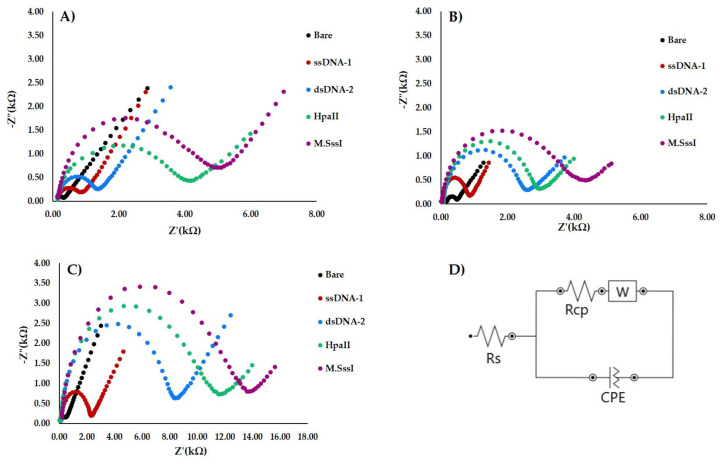
Electrochemical impedance spectroscopy (EIS) characterization of the electrodes prepared by different nanocomposites, where the Nyquist plot was recorded. (**A**) rGO-AgNPs, (**B**) rGO-CuNPs, (**C**) rGO-AuNPs and its (**D**) equivalent circuit.

**Figure 5 materials-13-04936-f005:**
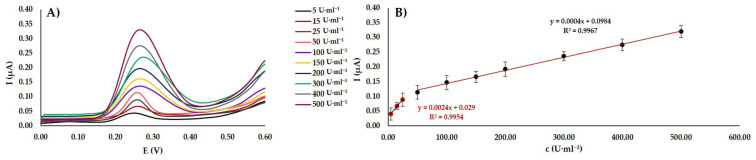
Monitoring of methyltransferase activity by differential pulse voltammetry (DPV). (**A**) The voltammograms of the modified WEs by ssDNA-1/dsDNA-2/rGO-AuNPs were then methylated by various concentrations of M.SssI Mtase. (**B**) The calibration curve showing the correlation between an electrochemical current response and concentration of M.SssI MTase. The error bars are represented using the relative standard deviation (RSD) of measurements (n = 3).

**Figure 6 materials-13-04936-f006:**
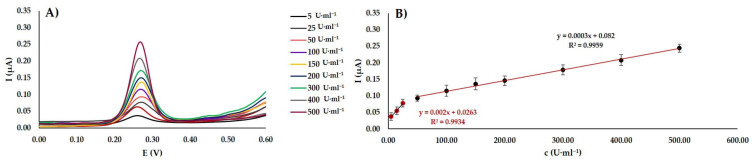
Analysis of methyltransferase activity by DPV in undiluted serum samples. (**A**) The voltammograms of the modified WEs by ssDNA−1/dsDNA-2/rGO-AuNPs were then methylated by various concentrations of M.SssI Mtase. (**B**) The calibration curve showing the correlation between an electrochemical current response and concentration of M.SssI MTase. The error bars are represented using the RSD% of measurements (n = 3).

**Table 1 materials-13-04936-t001:** Various study comparison in dependence of limit of detection (LOD) and nanocomposite platform used in the case of DNA methylation assays.

Nanocomposite Platform	Target Enzyme/Label	Technique	Linear Range [U·mL^−1^]	LOD [U·mL^−1^]	Reference
AuNPs electrodeposition	M.SssI MTase	DPV	0.50–400	0.04	[20]
DNA-AuNPs	Dam MTase	DPV	0.08–30	0.20	[28]
AuNPs electrodeposition	M.SssI MTase/FcA ^1^	DPV	0.50–400	0.10	[29]
AuNPs electrodeposition	T.aqI MTase	DPV	0.10–100	0.03	[30]
-	Dam MTase/MB	DPV	0.10–50	0.07	[31]
AuNPs electrodeposition	M.SssI MTase/HRP-IgG ^2^	DPV	0.05–80	0.10	[32]
-	Dam MTase/RuHex ^3^	DPV	0.25–10	0.18	[33]
AuNPs electrodeposition	M.SssI MTase	DPV	0.50–550	0.05	[34]
rGO-AuNPs	M.SssI MTase	DPV	50–500	0.06	This work

^1^ FcA—Ferrocenecarboxylic acid; ^2^ HRP-IgG—Horseradishperoxidase linked to secondary antibody; ^3^ RuHex—Hexaammineruthenium(III) chloride.

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
