# Peer review of "Label-Free DNA Biosensor Using Modified Reduced Graphene Oxide Platform as a DNA Methylation Assay"

_materials, 2020, doi:10.3390/ma13214936_

Round 1

Reviewer 1 Report

In this work, the authors presented a label-free DNA-based electrochemical biosensor made of nanoparticles, enzyme, and rGO.

The characterizations are comprehensive and the results look promising. I suggested that this paper is suitable to be published after some minor revision.

(1) Fo introduction, it lacks a clear rationale to describe the significance of this biosensor. Since this is a "material" journal, readers may not have in-depth cancer biology training. An introduction can help readers find the key-value of this work quicker and easier. Please include that in your revision.

(2) Ag nanoparticles are known to be toxic, please remove that statement in your introduction.

(3) In conclusion, the authors attribute high sensitivity can be due to the high electrical conductivity of AuNPs. Do authors consider other factors e.g. particle size, enzyme coverage/density of immobilized enzyme?     

Author Response

The Word file was uploaded.

Reviewer 2 Report

"Label-Free DNA Biosensor using Modified Reduced Graphene Oxide Platform as a DNA Methylation Assay" is an interesting research reporting on a  a new electrochemical sensing platform and its potential application in earlier diagnosis of DNA0methylation related pathological events. The biosensor was used with good results even in complex biological samples like serum.

To improve the quality of the article I suggest the followings:

  1. reformulation of the sentences: "The proposed scientific work reports..."; "The sensitivity of the biosensor was gradually
    enhanced by rGO decorated...";""Besides, this biosensor..."; "Therefore, various nanocomposites..."; "The usability of nanocomposites in biosensors ..." and "Mainly due to the more affordable materials..." ; "Other chemicals including methylene ..."; "Afterwards, EIS was used..."; "The immobilization of rGO-AuNPs with ssDNA-1 results in an increased ..."
  2. In section 2.3.2. "A volume of 1 ml of GO solution" is actually rGO not GO
  3. the caption of Fig. 2 *SEM of GO and rGO, should be corrected;
  4. in the legend of Fig. 4a, b, c, is better to replace "bar" by "rGO-AgNPs", "rGO-CuNPs"," rGO-AuNPs".
  5. the fig. 4D should be discussed in the article.
  6. in the Conclusions, the linerity range is actually 5-500 rGO-AgNPs
  7. In Supplementary material "caused by due to reduction" should be reformulated.
  8. It would be better to present in the Introduction other methods employed for DNA methylation assessment.

Author Response

The Word file was uploaded.

Reviewer 3 Report

This workk is dedicated to the development of a label-free DNA-based electrochemical biosensor which uses modified reduced graphene oxide decorated with gold, silver or copper nanoparticles. The papier is clear and well-written. The results are convincing and so I recomment publication of this paper after the following minor revisions: -l207: authors write: "SAM...enhances the electrochemical signal": I'm surprised by this asumption since a long carbon chain is not favorable to an enhancement of the electrochemical signal from my point of view, are you sure? have you any evidence to justify this asumption? -fig1: is Figure 1A really useful and meaningful? maybe it can be removed -fig3: can you explain the difference of size between AuNPs, AgNPs, CuNPs?? -fig5B and 6B: In the text we have the regression of the curve with a very good R2 and it is said that the range is going from 0 to 500 but on the graphs we see 2 lines which have very different slopes and which are both in the range 0-500. So I don't understand the correspondance between the text and the Figures, for me either it is not the correct detection range or there is two different regressions corresponding to this range -typographical errors: l107: please rephrase l149: "completereduction" l154: "H[AuCl4]" l159,160,166,197,244: write WEs instead of "WE's" l225: write "it is evident" l245: write "As it is shown" or "As shown" l265: "causes" not "cases"?? l293-294: rephrase, not clear  

Author Response

The Word file was uploaded.
